# Learning State-Aware Visual Representations from Audible Interactions

**Himangi Mittal**[1], **Pedro Morgado**[1], **Unnat Jain**[2], **Abhinav Gupta**[1]
[1]Carnegie Mellon University [2]Meta AI Research

## Abstract

We propose a self-supervised algorithm to learn representations from egocentric video data. Recently, significant efforts have been made to capture humans interacting with their own environments as they go about their daily activities. In result, several large egocentric datasets of interaction-rich multi-modal data have emerged. However, learning representations from videos can be challenging. First, given the uncurated nature of long-form continuous videos, learning effective representations require focusing on moments in time when interactions take place. Second, visual representations of daily activities should be sensitive to changes in the state of the environment. However, current successful multimodal learning frameworks encourage representation invariance over time. To address these challenges, we leverage audio signals to identify moments of likely interactions which are conducive to better learning. We also propose a novel self-supervised objective that learns from audible state changes caused by interactions. We validate these contributions extensively on two large-scale egocentric datasets, EPIC-Kitchens-100 and the recently released Ego4D, and show improvements on several downstream tasks, including action recognition, long-term action anticipation, and object state change classification. Code and pretrained model are available here: https://github.com/HimangiM/RepLAI

## 1 Introduction

Recent successes in self-supervised learning (SSL) [48, 10, 31, 28] has brought into question the need for human annotations in order to learn strong visual representations. However, current approaches are bottlenecked by the lack of rich data – they learn from static images which lack temporal information and restrict the ability to learn object deformations and state changes. It is clear that we need videos to learn rich representations in self-supervised manner.

Learning representations from videos is however quite challenging. The first challenge is choosing the right SSL loss. Approaches such as [67, 54] have attempted to learn representations that are invariant to object deformations/viewpoints. However, many downstream tasks require representations that are sensitive to these deformations. Another alternative has been to use the multi-modal data [3, 43, 57] and learn representations via audio. But again most of these approaches seek to align audio and visual features in a common space, leading to invariant representations as well. The second challenge is dealing with the fact that current video-based SSL approaches exploit the curated nature of video datasets, such as Kinetics [9]. These approaches are designed to leverage carefully selected clips, displaying a single action or object interaction. This is in contrast to the predominantly *untrimmed* real-world data characteristic of large egocentric datasets of daily activities. Here, unlike action centric datasets, the most 'interesting' or 'interaction-rich' clips have NOT been carefully selected by human annotators. Thus, learning from untrimmed video poses a major challenge, as a significant portion of the data does not focus on the concepts we want to learn.

36th Conference on Neural Information Processing Systems (NeurIPS 2022).

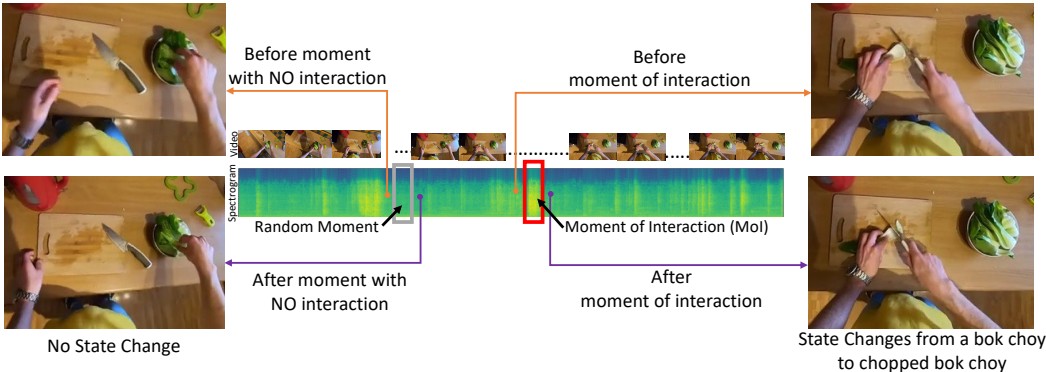

**Figure 1: Moments of audible interactions in a long, untrimmed video.** Random moments in time are likely to contain NO interactions, *e.g.*, timestamp shown by the gray box. Since no interactions occur, no changes in the before and after states are observed. By sampling from moments of interaction (MoI), as shown in the red box, we can learn representations that are sensitive to the change in the visual state caused by interactions.

In this work, we ask the question, 'Can we learn meaningful representations from interaction-rich, multi-modal streams of egocentric data?' Learning from continuous streams of data requires focusing on the right moments when the actual interactions are likely to occur. Consider, for example, the acts of opening a fridge or placing a pan on the stove. Actions like these create clear and consistent sound signatures due to the physical interaction between objects. These moments can be easily detected from audio alone and can be used to target training on interesting portions of the untrimmed videos. We show that even a simple spectrogram-based handcrafted detector is sufficient to identify interesting moments in time, and that representation learning benefits substantially from using them to sample training clips.

But what should the loss be? Prior work on audio-visual correspondence (AVC) [15, 4, 43] uses the natural co-occurrence of sounds and the visual manifestations of their sources as the source of supervision. However, since the AVC objective still favors invariance, the learned representations are not informative of the changes that happen over time (*e.g.*, representations that can distinguish between closed and opened fridge, or vegetables before and after chopping them). To better capture state changes, we introduce a novel audio-visual self-supervised objective, in which audio representations at key moments in time are required to be informative of the *change* in the corresponding visual representations over time. The intuition behind this objective is that transitions between object states are often marked by characteristic sounds. Thus, models optimized under this objective would associate the distinct sounds not only with the objects themselves (as accomplished with AVC), but also with the transition between two different states of the object.

To this end, we introduce RepLAI – **Rep**resentation **L**earning from **A**udible **I**nteractions, a self-supervised algorithm for representation learning from videos of audible interactions. RepLAI uses the audio signals in two unique ways: (1) to identify moments in time that are conducive to better self-supervised learning and (2) to learn representations that focus on the visual state changes caused by audible interactions. We validate these contributions extensively on two egocentric datasets, EPIC-Kitchens-100 [14] and the recently released Ego4D [27], where we demonstrate the benefits of RepLAI for several downstream tasks, including action recognition, long term action anticipation, and object state change classification.

## 2 Related Work

**Self-supervised learning.** Self-supervised learning methods operate on an unlabeled dataset by explicitly defining pretext tasks such as solving jigsaw puzzle [47], patch location prediction [16], inpainting [50], and image rotation [25] prediction. Following these, the next wave of self-supervised methods has been based on contrastive learning that learns representations with the help of data augmentation and instance discrimination [10, 28, 48, 31, 8]. These methods have shown rapid progress in self-supervised learning for images. While these approaches explore the spatial information of images, RepLAI leverages the temporal information of videos.

**Video representation learning.** Relevant to our proposed approach is self-supervised representation learning for videos where the spatiotemporal pretext tasks are designed such as temporal order prediction [40, 70, 35, 69], predicting motion and appearance statistics [65], pace prediction [66], temporal cycle consistency [18, 68], and video colorization [64]. Contrastive learning has also been widely adopted in the domain of video [55, 29, 32, 57, 71, 30, 22] with impressive results on action recognition tasks. These methods however learn representations that are invariant to spatio-temporal augmentations, such as temporal jittering, and thus are incapable of representing object state changes. Closer to the objective of RepLAI, we include relevant literature on audio-visual representation learning from videos, where the audio stream is additionally utilized.

**Audio-visual representation learning.** Learning without additional supervision has also been explored in the context of the audio modality with the help of audio-visual correspondence (AVC) [4, 5]. As stated simply, AVC is the binary classification task of predicting if a video clip and a short audio clip correspond with each other or not (details in Sec. 3.4). Similar tasks like temporal synchronization [36, 49] between audio and video, audio classification [6, 3, 11], spatial alignment prediction between audio and 360-degree videos [41], optimal combination of self-supervised tasks [52] have been shown beneficial for learning effective multi-modal video representations. Other works explore contrastive learning for both audio and video modality [43, 51, 42] as a cross-modal instance discrimination task.

**Fine-grained video understanding.** Real-world videos are often untrimmed in nature and have multiple actions in a single video. Along this line, fine-grained analysis has been studied for videos in the form of a query-response temporal attention mechanism [72], bi-directional RNN s[58], and semi-supervised learning problem [17]. While these works only utilize the visual modality, another line of work has also explored multi-modal fine-grained video understanding as a transformer-based model [34], by exploiting the correspondence between modalities [44], or by exploring how to best combine multiple modalities - audio, visual, and language [2]. In our work, we try to conduct fine-grained video understanding in a self-supervised manner.

**Egocentric datasets.** Egocentric datasets offer new opportunities to learn from a first-person point of view, where the world is seen through the eyes of an agent. Many egocentric datasets have been developed such as Epic-kitchens [13, 14] which consist of daily activities performed in a kitchen environment, Activities of Daily Living [53], UT Ego [37, 60], the Disney Dataset [20], and the recently released large-scale Ego4D dataset [27] which consists of day-to-day life activities in multiple scenarios such as household, outdoor spaces, workplace, etc. Multiple challenges and downstream tasks have explored for egocentric datasets like action recognition [34, 33, 38], action localization [56], action anticipation [26, 59, 39, 1, 23], human-object interactions [45, 12, 7], parsing social interactions [46], and domain adaptation [44]. In our work, we evaluate the efficiency of the representations learned by our self-supervised approach on the EPIC-Kitchens-100 and Ego4D datasets over multiple downstream tasks.

## 3 RepLAI

In this section, we detail our approach to learn audio-visual representations from and for interaction-rich egocentric data in a self-supervised manner, *i.e.*, without relying on human annotated labels. Sec. 3.1 provides an overview of RepLAI and motivates the two key contributions of this work – identifying 'moments of interaction' (MoI) and learning from 'audible visual state changes'. Sec. 3.2 details the proposed approach for MoI detection and section Sec. 3.3 explains the proposed self-supervised objective for learning state-aware representations. Sec. 3.4 explains the objective of audio-visual correspondence learning used to train RepLAI. Sec. 3.5 brings both objectives together and includes necessary details for reproducibility.

### 3.1 Overview

Given a dataset $\mathcal{D} = \{(v_i, a_i)_{i=1}^N\}$ containing $N$ long (untrimmed) audio-visual streams, our goal is to learn visual and audio encoders, denoted $f_V$ and $f_A$, that can effectively represent egocentric data. An overview of the proposed approach is depicted in Fig. 2. For each sample $(v, a) \in \mathcal{D}$, we search for moments of interaction (MoI) using the audio stream, and extract short audio and visual clips around these MoI. These trimmed clips are then encoded into a vectorized representation using $f_V$ and $f_A$. The whole system is trained to optimize two self-supervised losses – an audio-visual

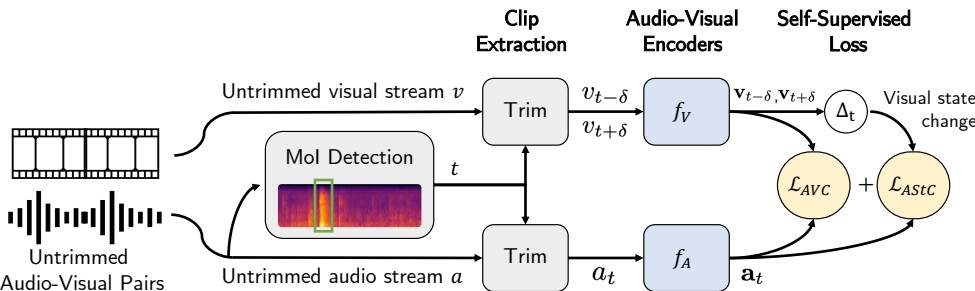

**Figure 2: Overview of RepLAI.** RepLAI seeks to learn audio and visual encoders ($f_A$ and $f_V$) by (1) detecting and focusing training data on moments of interaction (MoI) present in untrimmed videos and (2) solving a combination of two tasks – audio-visual correspondence (`AVC`) and audio identifiable state changes (`AStC`).

correspondence loss $\mathcal{L}_{\texttt{AVC}}$, and a novel self-supervised loss that learns from audible state changes $\mathcal{L}_{\texttt{AStC}}$.

*Why detect moments of interaction (MoI)?* Untrimmed video of daily activities often contains long periods without interactions, which aren't useful for training. Instead, we search for moments in time that are more likely to contain interactions which we refer to as moments of interaction (MoI).

*Why learn from audible state changes?* Visual representations of daily activities should be informative of the state of the environment and/or objects being interacted with. Moreover, changes in the environment are usually caused by physical interactions, which produce distinct sound signatures. We hypothesize that state-aware representations can be obtained by learning to associate audio with the change of visual representation during a moment of interaction.

## 3.2  Audio-driven detection of moments of interaction

Audio signals are particularly informative of moments of interaction. To complete day-to-day activities, we physically interact with objects in our environments. These interactions typically produce distinct audio patterns - short bursts of energy that span all frequencies. This is illustrated in Fig. 1, where we visualize the untrimmed visual and audio data of a person performing a series of actions in the kitchen. The audio data is represented as a log mel spectrogram, where the x-axis represents time and y-axis the audio frequency in log-scale. As can be seen, moments of interaction appear in the spectrogram as *vertical edges*, which can be easily detected. Once detected, short clips around the moments of interaction are collected into a dataset $\mathcal{D}_{\texttt{MoI}}$, and used for training.

The remaining question is *how to locate the timestamp of such vertical edges?* Intuitively, we do this by finding robust local maxima in the total energy (summed over all frequencies) of the spectrogram. Concretely, let $M(t, \omega)$ be the value of the log mel spectrogram of an audio clip at time $t$ and frequency $\omega$. To remove the influence of background noise and overall audio intensity/volume, we compute the z-score normalization of the spectrogram for each frequency independently $\bar{M}(t, \omega) = \frac{s(t, \omega) - \mu_\omega}{\sigma_\omega + \epsilon}$, where $\epsilon$ is small constant for numerical stability. Here, $\mu_\omega$ and $\sigma_\omega$ are the mean and standard deviation of $M(t, \omega)$ over time, respectively.[1] Next, we define moments of interaction as the set of timestamps which are local maxima of $\sum_\omega \bar{s}(t, \omega)$ (or peaks for short). Moreover, to avoid weak local maxima that may be caused by the noisy nature of audio signals, we ignore peaks with small prominence (lower than 1)[2]. For further robustness, when multiple close peaks are found (less than 50ms apart), only the highest prominence peak is kept.

## 3.3  Learning from audible state changes

Physical interactions often cause both state changes in the environment and distinct audio signals. To leverage this natural co-occurrence, we propose a self-supervised task that seeks to *associate the audio with changes in the visual state* during a moment of interaction.

---

[1]Specifically, $\mu_\omega = \mathbb{E}_t[M(t, \omega)]$, $\sigma_\omega^2 = \mathbb{E}_t[(M(t, \omega) - \mu_\omega)^2]$, and $\epsilon = 1e - 5$.
[2]The prominence of a peak is defined as the difference between the peak value and the minimum value in a small window around it.

**(a)** The proposed `AStC` formulation (Sec. 3.3)     **(b)** Audio-visual correspondence (Sec. 3.4)

**Figure 3: RepLAI architecture for `AStC` and `AVC` tasks.** In both cases, short clips are extracted around moments of interaction (MoI). (a). In `AStC`, the representation of a visual state change $\Delta\mathbf{v}_t$ is matched to the corresponding audio $\mathbf{a}_t$. (b). `AVC` seeks to associate audio $a_t$ with the corresponding visual clips $v_t$.

The proposed task is optimized by minimizing a loss with two negative log-likelihood terms to: (1) *increase* the probability of associating the audio with the visual state change in the *forward* (*i.e.* correct) direction, (2) *decrease* the probability of associating the audio with the visual state change in the *backward* (*i.e.* incorrect) direction. Consider, for example, the interaction of 'closing a fridge door'. To optimize for this task, the audio of closing the door should be (1) similar to the visual transition *opened door → closed door* and (2) dissimilar to the (backwards) transition *closed → open*. This encourages learning of representations that are informative of object states, making them useful for a variety of egocentric tasks. Specifically, the audible state change (`AStC`) loss is defined as

$$\mathcal{L}_{\texttt{AStC}} = \mathbb{E}_{v_t, a_t \in \mathcal{D}_{\texttt{MoI}}} \left[ -\log\left(p^{\text{frwd}}(v_t, a_t)\right) - \log\left(1 - p^{\text{bkwd}}(v_t, a_t)\right) \right]. \tag{1}$$

The probabilities ($p^{\text{frwd}}$, $p^{\text{bkwd}}$) are computed from cross-modal similarities

$$p^{\text{frwd}}(v_t, a_t) = \sigma\left(\texttt{sim}\left(\Delta\mathbf{v}_t^{\text{frwd}}, \mathbf{a}_t\right)/\tau\right), \tag{2}$$

$$p^{\text{bkwd}}(v_t, a_t) = \sigma\left(\texttt{sim}\left(\Delta\mathbf{v}_t^{\text{bkwd}}, \mathbf{a}_t\right)/\tau\right), \tag{3}$$

where $\tau = 0.2$ is a temperature hyper-parameter, and $\sigma$ denotes the sigmoid function. For better readability, we absorb the notations for the audio projection MLP head $h_A^{\texttt{AStC}}$ and the state change projection MLP head $h_{\Delta V}^{\texttt{AStC}}$ within $\texttt{sim}(\cdot, \cdot)$, but their usage is clearly illustrated in Fig. 3a.

**Audio representations ($\mathbf{a}_t$)** are obtained by encoding the trimmed audio clips $a_t$ via the audio encoder $f_A$ (shared across all objectives). As explained above, $\mathbf{a}_t$ is further projected via $h_A^{\texttt{AStC}}$ to a space where similarity to visual state changes is enforced.

**State change representations ($\Delta\mathbf{v}_t^{\textbf{frwd}}$, $\Delta\mathbf{v}_t^{\textbf{bkwd}}$)** are computed by considering two non-overlapping visual clips for each moment of interaction $t$, at timestamps $t - \delta$ and $t + \delta$. The two clips, $v_{t-\delta}$ and $v_{t+\delta}$, are encoded via the visual encoder $f_V$ (shared across all tasks) and a projection MLP head $h_V^{\texttt{AStC}}$ (specific to the `AStC` task). Specifically, we represent forward and backward state changes as

$$\Delta\mathbf{v}_t^{\text{frwd}} = h_V^{\texttt{AStC}} \circ f_V(v_{t+\delta}) - h_V^{\texttt{AStC}} \circ f_V(v_{t-\delta}), \tag{4}$$

$$\Delta\mathbf{v}_t^{\text{bkwd}} = h_V^{\texttt{AStC}} \circ f_V(v_{t-\delta}) - h_V^{\texttt{AStC}} \circ f_V(v_{t+\delta}). \tag{5}$$

In summary, optimizing the loss of Eq. 1 not only requires the audio representation $\mathbf{a}_t$ to be aligned with representation of the visual change $\Delta\mathbf{v}_t^{\text{frwd}}$ that took place, but also to be different from the hypothetical backward state change $\Delta\mathbf{v}_t^{\text{bkwd}}$.

### 3.4 Learning from audio-visual correspondences [15, 4, 43]

Audio-visual correspondence (`AVC`) is a well-studied self-supervised methodology for learning unimodal audio and visual encoders. The key idea is to bring visual and audio clips into a common feature space, where the representations of audio-visual pairs are aligned. Note that `AVC` differs from the proposed `AStC` task, as `AVC` seeks to associate the audio $a_t$ with the corresponding visual clips $v_t$, as opposed to the change in visual state $\Delta\mathbf{v}_t$. As a result, visual representations learned through `AVC` are biased towards static concepts, while those learned through `AStC` are more sensitive to dynamic

concepts. Since both types of representations can be useful for egocentric tasks, we further train the visual and audio encoders, $f_V$ and $f_A$, for the AVC task.

Specifically, consider a dataset of audio-visual pairs $(v_i, a_i)$ with representations $\mathbf{v}_i = f_V(v_i)$ and $\mathbf{a}_i = f_A(a_i)$. In particular, we let $(v_i, a_i)$ be short clips extracted from sample $i$ around one of the detected moments of interest. Then, following [43, 61], audio-visual correspondence is established by minimizing a cross-modal InfoNCE loss of the form

$$\mathcal{L}_{\texttt{AVC}} = \mathbb{E}_{v_i, a_i \sim \mathcal{D}} \left[ -\log \frac{e^{\texttt{sim}(\mathbf{v}_i, \mathbf{a}_i)/\tau}}{\sum_j e^{\texttt{sim}(\mathbf{v}_i, \mathbf{a}_j)/\tau}} - \log \frac{e^{\texttt{sim}(\mathbf{v}_i, \mathbf{a}_i)/\tau}}{\sum_j e^{\texttt{sim}(\mathbf{v}_j, \mathbf{a}_i)/\tau}} \right], \tag{6}$$

where $\tau = 0.07$ is a temperature hyper-parameter and $\texttt{sim}(\cdot, \cdot)$ denotes the cosine similarity. Both terms in Eq. 6 help bring $\mathbf{v}_i$ and $\mathbf{a}_i$ (*i.e.* the positives) together. The key difference is whether the negative set is composed of audio representations $\mathbf{a}_j$ or visual representations $\mathbf{v}_j$ where $j \neq i$

For readability of Eq. 6, we once again absorb the notation for the audio and visual projection MLP heads ($h_A^{\texttt{AVC}}$ and $h_V^{\texttt{AVC}}$) within $\texttt{sim}(\cdot, \cdot)$, and illustrate their usage in Fig. 3b. Fig. 3b also shows that we apply the AVC loss twice to associate both the visual clips (extracted slightly before and after the moment of interaction $t$) to the corresponding audio.

## 3.5 Training

The audio-visual representation models $f_A$ and $f_V$ are trained to minimize both AVC and AStC losses

$$\mathcal{L} = \alpha \mathcal{L}_{\texttt{AVC}} + (1 - \alpha) \mathcal{L}_{\texttt{AStC}} \tag{7}$$

where $\alpha$ is a weighting hyper-parameter between the two terms. While we experimented with different values of $\alpha$, we found that equal weighting produced best results.

**Implementation details**. We follow prior work on audio visual correspondence [43], and use an R(2+1)D video encoder [62] with depth 18 and a 10-layer 2D CNN as the audio encoder. Two video clips are extracted around moments of interaction at a frame rate of 16 FPS each with a duration of 0.5s, and separated by a gap of 0.2s. Video clips are augmented by random resizing, cropping, and horizontal flipping resulting in clips of 8 frames at a resolution of $112 \times 112$,. As for the audio, we extract clips of 2s at 44.1kHz and downsample them to 16kHz. If the audio is stereo, we average the two waveforms to downgrade to mono, and then convert the mono signal to a log mel spectrogram with 80 frequency bands and 128 temporal frames. Models are trained with stochastic gradient descent for 100 epochs with a batch size of 128 trained over 4 GTX 1080 Ti GPUs, a learning rate of 0.005 and a momentum of 0.9. For Ego4D, we use a batch size of 512 trained over 8 RTX 2080 Ti GPUs with a learning rate of 0.05. The two loss terms in Eq. 7 are equally weighted with $\alpha = 0.5$.

# 4 Experiments

In this section, we demonstrate the benefits of identifying moments of interaction and learning state-aware representations through an audible state-change objective. We also show that, while large scale audio-visual correspondence (AVC) is beneficial, it is not sufficient to learn state-aware representations required for egocentric tasks. The setup used for our experiments is described in Sec. 4.1. Results and discussion of main takeaways are presented in Sec. 4.2.

## 4.1 Experimental Setup

**Datasets**. We evaluate on two egocentric datasets: EPIC-Kitchens-100 [14] and Ego4D [27]. EPIC-Kitchens-100 contains 100 hours of activities in the kitchen. Ego4D contains 3670 hours of egocentric video covering daily activities in the home, workplace, social settings, *etc*. For experiments on Ego4D, we use all videos from the Forecasting and Hand-Object interaction subsets.

**Baselines and ablations.** We consider various baselines as well as ablated versions of RepLAI. *Random* represents an untrained (randomly initialized) model. *AVID* [43] and *XDC* [3] are two state-of-the-art models pre-trained on 2M audio-visual pairs from AudioSet [24] that only leverage audio-visual correspondence. For the full method *RepLAI*, we initialize the model weights from AVID before training on moments of interaction to minimize both AVC and state change loss, AStC.

| Method | $\mathcal{L}_{\text{AVC}}$ | $\mathcal{L}_{\text{AStC}}$ | MoI Sampling | AVC Pretraining [43] | Top1 Acc ↑ Verb | Top1 Acc ↑ Noun | Top5 Acc ↑ Verb | Top5 Acc ↑ Noun |
|---|---|---|---|---|---|---|---|---|
| (1) Random | | | | | 20.38 | 4.96 | 64.75 | 19.83 |
| (2) XDC [3] | | | | | 24.46 | 6.75 | 68.04 | 22.71 |
| (3) AVID [43] | | | | ✓ | 26.62 | 9.00 | 69.79 | 25.50 |
| (4) RepLAI w/o AVC | | ✓ | ✓ | ✓ | 29.92 | 10.46 | 70.58 | 29.00 |
| (5) RepLAI w/o AStC | ✓ | | ✓ | ✓ | 29.29 | 9.67 | 73.33 | 29.54 |
| (6) RepLAI w/o MoI | ✓ | ✓ | | ✓ | 28.71 | 8.33 | 73.17 | 27.29 |
| (7) RepLAI (scratch) | ✓ | ✓ | ✓ | | 25.75 | 8.12 | 71.25 | 27.29 |
| (8) RepLAI | ✓ | ✓ | ✓ | ✓ | **31.71** | **11.25** | **73.54** | **30.54** |

**Table 1:** Action recognition on EPIC-Kitchens-100. Top1 and top5 accuracy (%) is reported. ↑: Higher is better.

We also evaluate our method trained without AVID initialization (*RepLAI from scratch*), trained with only AVC (*RepLAI w/o AStC*), only state change losses (*RepLAI w/o AVC*), and trained on random moments in time (*RepLAI w/o MoI*). Finally, we compare our approach with the fully supervised methods presented in Ego4D [27].

**Downstream tasks.** After self-supervised pre-training, the models are evaluated on a range of egocentric downstream tasks. This is done, as is standard, by appending a task specific decoder to the backbone model, and training the decoder on a small annotated dataset. The tasks are:

- *Video action recognition (AR) on EPIC-Kitchens-100 and Ego4D.* Given a short video clip, the task is to classify the 'verb' and 'noun' of the action taking place. This is done using two separate linear classifiers trained for this task. We report the top-1 and top-5 accuracies, following [14] (Tab. 1) and [27] (Tab. 2). We also evaluate on the unseen participants, head classes, and tail classes of EPIC-Kitchens-100 in Tab. 3. Through this task, we assess the efficacy of the spatial-temporal representations learned by the model in differentiating among different verbs and nouns.

- *Long-term action anticipation (LTA) on Ego4D.* Given a video, the task is to predict the camera wearer's future sequence of actions. For this task, the model is first presented with 4 consecutive clips of 2s, which are encoded using our visual backbone $f_V$. Following [27], the representations are concatenated and fed to 20 separate linear classification heads to predict the following 20 actions. Performance is measured using the edit distance metric ED@(Z=20) [27].[3] With the help of this task, we can evaluate if the representations learned by the model can be employed for long-horizon planning where the actions can change and may be of arbitrary duration. Results are reported in Tab. 2.

- *State change classification (StCC) on Ego4D.* Given a video clip, the task is to classify if an object undergoes a state change or not. The video clip is encoded by $f_V$ and a state change classification head is used which performs global average pooling on the entire feature tensor and is followed by a classification layer. Performance is measured through the State Change Classification Accuracy (%), and reported in Tab. 2. This task is ideal for assessing the ability of the model in understanding the temporal change happening in the state of an object.

### 4.2 Discussion of results

As can be seen in Tab. 1 and Tab. 2, RepLAI outperforms all other methods across all downstream tasks. Overall, this can be attributed to its ability to focus on interactions, both by detecting when they occur and by learning representations that are sensitive to interactions. A closer analysis of these results reveals several insights that we discuss next.

**RepLAI enhances large-scale AVC driven approaches.** Prior work on self-supervised audio-visual learning has shown strong audio-visual representations for action recognition [43, 42]. One question that we seek to answer is, how useful these are representations to egocentric tasks and what are their limitations? To answer this question, we compare our model trained from scratch, *RepLAI (Scratch)*,

---

[3]Edit distance measures the minimum number of operations required to convert the predicted sequence of actions to ground truth. To account for multi-modality of future actions, it also allows the model to make $Z = 20$ predictions, and only accounts for the best prediction.

| Method | $\mathcal{L}_{\texttt{AVC}}$ | $\mathcal{L}_{\texttt{AStC}}$ | MoI | AVC Pretraining [43] | StCC Acc ↑ | AR Top1 Acc ↑ Verb | AR Top1 Acc ↑ Noun | LTA ED@(Z=20) ↓ Verb | LTA ED@(Z=20) ↓ Noun |
|---|---|---|---|---|---|---|---|---|---|
| (S1) I3D-ResNet-50 [9, 27] | NA | NA | NA | NA | 68.70 | - | - | - | - |
| (S2) SlowFast [21, 27] | NA | NA | NA | NA | - | - | - | 0.747 | 0.808 |
| (S3) MViT [19, 27] | NA | NA | NA | NA | - | - | - | 0.707 | 0.901 |
| (1) Random | | | | | 51.80 | 17.4 | 7.7 | 0.831 | 0.936 |
| (2) XDC [3] | | | | | 58.90 | 17.90 | 8.70 | 0.823 | 0.928 |
| (3) AVID [43] | | | | ✓ | 61.11 | 18.3 | 10.7 | 0.811 | 0.919 |
| (4) RepLAI w/o AVC | | ✓ | ✓ | ✓ | 64.00 | 20.3 | 12.4 | 0.781 | 0.854 |
| (5) RepLAI w/o AStC | ✓ | | ✓ | ✓ | 63.60 | 21.1 | 13.5 | 0.774 | 0.853 |
| (6) RepLAI w/o MoI | ✓ | ✓ | | ✓ | 62.90 | 19.8 | 11.2 | 0.792 | 0.868 |
| (7) RepLAI (scratch) | ✓ | ✓ | ✓ | | 66.20 | 22.2 | 14.1 | 0.760 | 0.840 |
| (8) RepLAI | ✓ | ✓ | ✓ | ✓ | **66.30** | **22.5** | **14.7** | **0.755** | **0.834** |

**Table 2:** Performance on several downstream tasks on Ego4D. StCC: State Change Classification (%). AR: Action Recognition (%). LTA: Long-term action anticipation. ↑: Higher is better. ↓: Lower is better.

| Methods | Unseen Participants Top1 Acc ↑ Verb | Unseen Participants Top1 Acc ↑ Noun | Unseen Participants Top5 Acc ↑ Verb | Unseen Participants Top5 Acc ↑ Noun | Tail Classes Top1 Acc ↑ Verb | Tail Classes Top1 Acc ↑ Noun | Tail Classes Top5 Acc ↑ Verb | Tail Classes Top5 Acc ↑ Noun | Head Classes Top1 Acc ↑ Verb | Head Classes Top1 Acc ↑ Noun | Head Classes Top5 Acc ↑ Verb | Head Classes Top5 Acc ↑ Noun |
|---|---|---|---|---|---|---|---|---|---|---|---|---|
| XDC [3] | 24.29 | 6.96 | 67.79 | 23.00 | 15.89 | 4.17 | 44.92 | 9.77 | 24.78 | 6.95 | 72.28 | 24.74 |
| AVID [43] | 26.17 | 8.67 | 68.75 | 24.12 | 16.80 | 4.43 | 47.14 | 12.89 | 27.95 | 9.82 | 73.20 | 28.43 |
| RepLAI w/o AVC | 28.67 | 9.38 | 72.04 | 27.88 | 18.49 | 5.21 | 47.79 | 12.63 | 30.59 | 10.21 | 73.33 | 30.90 |
| RepLAI w/o MoI | 27.71 | 7.92 | 72.08 | 26.88 | 16.80 | 4.04 | 49.74 | 12.76 | 29.36 | 10.65 | 76.33 | 30.41 |
| RepLAI | **31.58** | **10.17** | **73.46** | **29.96** | **20.05** | **6.12** | **52.08** | **16.54** | **33.41** | **11.58** | **77.77** | **34.33** |

**Table 3:** Video action recognition (AR) accuracy (%) on EPIC-Kitchens-100 for unseen participants, head classes, and tail classes. Top1 and top5 accuracy (%) is reported. ↑: Higher is better.

with our model using the weights from AVID [43] as initialization for both the visual and audio encoders. We also compare our method to standalone AVID and XDC *i.e.* without further self-supervised training. Comparing rows (2), (3) and (8) in Tab. 1 and Tab. 2, it is clear that RepLAI enhances large-scale AVC pre-training by significant margins, leading to absolute improvements of 5% in top-1 verb accuracy on EPIC-Kitchens-100, 4.2% on Ego4D, 5.2% increase in state-change classification accuracy, and 5.6% reduction on the edit distance for long-term anticipation compared to AVID. Comparing rows (7) and (8), we also see that large-scale AVID pre-training enhances the representations learned by RepLAI on EPIC-Kitchens-100 significantly but only marginally on Ego-4D. This is likely due to the significantly large diversity of scenes in Ego4D. Thus, while relying on large-scale audio-visual pre-training (as with AVID) can help avoid overfitting on smaller egocentric datasets, this is less critical when training on larger and more diverse data.

**Detecting moments of interaction (MoI) helps representation learning.** We hypothesize that to learn good representations for egocentric data of daily activities, self-supervised learning should focus on moments in time when interactions occur. To assess whether our audio-driven MoI detection algorithm helps representation learning, we compare *RepLAI* with an ablated version, *RepLAI w/o MoI*, where the model is trained on audio-visual clips extracted at *random* from the untrimmed videos. As can be seen by comparing rows (6) and (8) in Tab. 1 and Tab. 2, sampling clips around MoI leads to significantly better representations for all egocentric downstream tasks that we study. Moreover, even though *RepLAI w/o MoI* trains with AStC, it is unable to fully leverage the state change objective function without the information of moments of interactions which leads to a worse performance. This suggests that, an explicit state change objective function and sampling video clips around moments of interactions (which are likely to be aligned with the actual state changes) together provide an information-rich feedback to our model in better understanding how the state changes by an interaction and how the actions transition over time. These results also clearly show that the proposed MoI detection procedure is able to find moments in time that are especially useful for learning representations of daily activities. We emphasize the simplicity and effectiveness of our audio-driven detector, which shows how informative audio can be when searching for moments of interaction. In the future, we believe that learning-based approaches could further enhance MoI detection, and further improve the learned audio-visual representations. We also show several qualitative examples of detected MoI in the supplement.

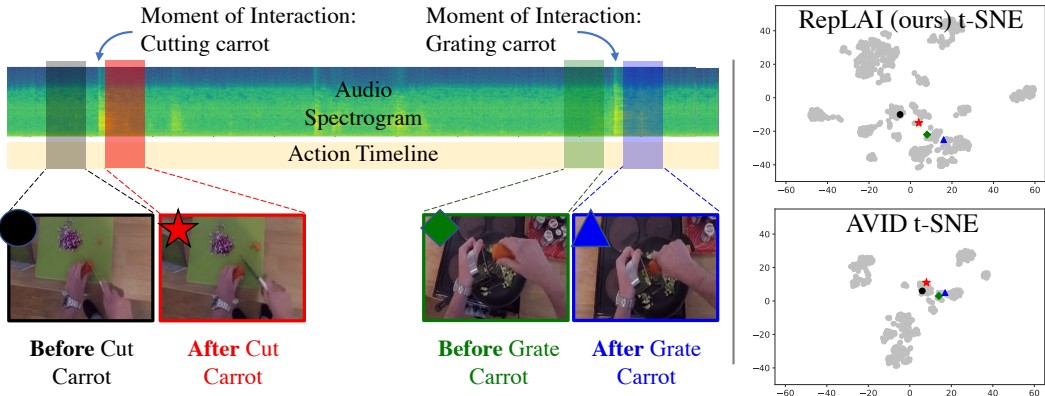

**Figure 4:** t-SNE visualization of the feature representations learned by *RepLAI* and *AVID* for a video consisting of fine-grained actions over time. For a simpler visualization, we consider all the videos belonging to a single participant. A larger spread in the t-SNE of *RepLAI* indicates more distinct state-aware representations.

`AVC` **and** `AStC` **are complementary.**     To assess the impact of both terms in Eq. 7, we evaluate *RepLAI* trained without $\mathcal{L}_{\texttt{AVC}}$ and without $\mathcal{L}_{\texttt{AStC}}$. Comparing rows (4), (5) to row (2) and row (3) in Tab. 1 and Tab. 2 shows that each term enhances the representations obtained through large-scale audio-visual pre-training (AVID). Furthermore, comparing the ablated models in rows (4) and (5) to the full model in row (8) shows that these two terms are complementary to each other. This is because the `AVC` and `AStC` tasks encourage learning of representations with different characteristics. `AVC` focuses on learning visual representations that are informative of what kind of sounding objects are present in the video, while `AStC` forces the model to differentiate between visual representations that occur before and after state change interactions.

**RepLAI encourages state-aware representation learning.**     To study the representations learned by our approach for different states, we generate a t-SNE plot [63] for *RepLAI* and *AVID* as shown in Fig. 4. For generating a simpler visualization, a small dataset is prepared consisting of all the videos corresponding to a single participant, *P01*, in EPIC-Kitchens-100 and split into clips of 0.5s. We can observe that there is a larger spread in the t-SNE plot for *RepLAI* than *AVID*. A larger spread indicates that the representations of the various states are significantly different from each other and form more distant clusters as shown by *RepLAI*. Whereas, if the state representations are similar to each other, they are clustered together and show lesser spread as shown by *AVID*. MoI are the key moments of interactions with an object in an environment where the state is changing. *AVID* has no such information about the key moments and also does not have an explicit state change objective function. Therefore, it is unable to discriminate between the *before* and *after* state of an action and has less effective state-aware information in its representations.

**RepLAI representation are more generalizable and robust to long-tail.**     To assess RepLAI in a scenario with domain shift, we evaluate on unseen participants that were fully excluded from the pre-training of RepLAI. Tab. 3 shows that RepLAI significantly outperforms baselines and ablations, indicating that representation learning by our model provides much better generalization. Moreover, the verb and noun classes in EPIC-Kitchens-100 exhibit a long-tailed distribution. When further compared on head and tail classes separately in Tab. 3, we can observe that RepLAI outperforms all other methods highlighting its higher robustness on a long-tailed distribution.

**Self-supervised vs supervised representation learning**     Tab. 2 also compares RepLAI to fully supervised methods introduced in Ego4D [27] (rows S1, S2 and S3). We can observe that RepLAI can also perform competitively to the fully supervised approaches when we have access to larger and more diverse data. With further focus on SSL for untrimmed datasets, SSL methods will be able to match supervised approaches, and our work takes a step towards it.

# 5  Conclusion

In this work, we propose an audio-driven self-supervised method for learning representations of egocentric video of daily activities. We show that in order to learn strong representations for this domain, two important challenges need to be addressed. First, learning should focus on moments of interaction (MoI). Since these moments only occur sporadically in untrimmed video data, we show that MoI detection is an important component of representation learning in untrimmed datasets. Second, learning should focus on the consequences of interactions, *i.e.*, changes in the state of an environment caused by agents interacting with the world. In particular, by seeking to identify visible state changes from the audio alone, we can learn representations that are potentially more aware of the state of the environment and hence, particularly useful for egocentric downstream tasks.

**Acknowledgements**

We would like to thank DARPA MCS, ONR Young Investigator and DARPA SAIL-ON for the funding.

**Broader impact**

Deep learning models are capable of learning (and sometimes even amplifying) biases existing in datasets. While several steps have been taken in datasets like Ego4D to increase geographical diversity, we would like to encourage careful consideration of ethical implications when deploying these model. While public datasets are essential to make progress on how to represent visual egocentric data, premature deployment of our models is likely have negative societal impact, as we did not check for the presence or absence of such biases.

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
