# Appendix

## A  Additional downstream evaluation tasks

We evaluated all models on three additional tasks, beyond those presented in the main paper. Following [3], we also evaluated models on localizing the point-of-no-return in state changes. We also assess audio representations by considering two multi-modal downstream tasks, namely, state change classification and action recognition.

*Point-of-no-return (PNR) temporal localization error:* Given a video clip of a state change, the network has to estimate the time at which a state change begins. More specifically, the model tries to estimate the keyframe within the video clip that contains the point-of-no-return (the time when the state change begins). This is done by training a fully-connected head applied to each frame's representation in order to identify the timestamp at which the state of an object changes. Performance is measured using temporal localization error (seconds) in Tab. S1.

*State change classification (StCC) w/ audio:* For this task, representations from both video and audio modalities are concatenated. The occurrence of state change is then predicted by training a binary linear classifier, using the concatenated representations as input. Performance is measured using state-change classification accuracy (%).

*Action Recognition (AR) w/ audio:* For this task, video embeddings from $f_V$ and audio embedding from $f_A$ are concatenated together and passed through two separate linear classifiers to classify the 'verb' and 'noun' of the action occurring in the video clip. Performance is measured using classification accuracy (%).

| Method | $\mathcal{L}_{\texttt{AVC}}$ | $\mathcal{L}_{\texttt{AStC}}$ | MoI | AVC Pretraining [5] | PNR Err ↓ | StCC w/ audio Acc ↑ | AR w/ Audio Top1 Acc ↑ Verb | AR w/ Audio Top1 Acc ↑ Noun |
|---|---|---|---|---|---|---|---|---|
| (S1) I3D-ResNet-50 [2, 3] | NA | NA | NA | NA | 0.739 | - | - | - |
| (S2) BMN [3, 4] | NA | NA | NA | NA | 0.780 | - | - | - |
| (1) Random | | | | | 0.827 | 52.90 | 18.90 | 9.50 |
| (2) XDC [1] | | | | | 0.820 | 57.70 | 19.10 | 10.20 |
| (3) AVID [5] | | | | ✓ | 0.814 | 61.30 | 19.80 | 12.30 |
| (4) RepLAI w/o AVC | | ✓ | ✓ | ✓ | 0.792 | 64.60 | 22.70 | 14.00 |
| (5) RepLAI w/o AStC | ✓ | | ✓ | ✓ | 0.795 | 64.40 | 21.40 | 13.00 |
| (6) RepLAI w/o MoI | ✓ | ✓ | | ✓ | 0.801 | 64.10 | 20.80 | 11.70 |
| (7) RepLAI (scratch) | ✓ | ✓ | ✓ | | 0.775 | 66.30 | 22.50 | 15.00 |
| (8) RepLAI | ✓ | ✓ | ✓ | ✓ | **0.772** | **66.80** | **23.10** | **15.80** |

**Table S1:** Performance on several downstream tasks on Ego4D. StCC: State Change Classification (%). PNR: Point-of-no-return temporal localization error (seconds). AR: Action Recognition (%). ↑: Higher is better. ↓: Lower is better.

**Discussion of Results**  The results on the additional downstream tasks are shown in Tab. S1. Point-of-no-return (PNR) temporal localization error shows that RepLAI enhances large-scale AVC pre-training significantly (as seen by comparing rows (2), (3) and (8)), moment of interaction (MoI) helps representation learning (comparing rows (6) and (8)), and finally, AVC and AStC are complementary when comparing rows (4), (5) and (8). Additionally, by comparing Tab. S1 with Table 2 in the main paper, we observe that the performance on state change classification (StCC) and action recognition (AR) improves by incorporating the audio modality, thus showing the usefulness of audio representations. The gains of incorporating audio can be seen across all models, but are more significant on action recognition.

## B  Analysis of RepLAI potential failure modes

To provide further insights onto the generalization ability of the proposed method, we conduct an experiment to assess how discriminative the learned representations are for different types of activities. For this experiment, we first categorize the activities based on the nature of the transition:

**T1:** irreversible interactions, backward transition highly unlikely (e.g., cut vegetables)

**T2:** reversible interactions, backward transition occurs often (e.g., open/close fridge)

**T3:** interactions with no transition direction (e.g., stirring).

`AStC` learns from both T1 and T2 interactions, as they are associated with visual state changes. Although T1 interactions are never seen in reverse order, the model still benefits from knowing the correct order, as this leads to more state-aware representations. As for T3 type interactions, they can be a failure mode of the `AStC` objective, if they cause no change in the visual state of the environment.

### B.1    Generalization and state change identifiability

To analyse how RepLAI representations behave for different types of interaction, we show several metrics in Tab. S2. We computed the mean average precision, after training a linear classifier for action recognition on Epic-Kitchens. The results indicate the RepLAI performs significantly better than the finetuned AVID baseline across all categories of transition/direction, showing that RepLAI (which includes both AVC and AStC) is generic enough to enhance representations for all types of interactions.

We also observed that MoI detection helps finding timestamps that have more perceptible visual state change (even for T3 type interactions). To see this, we computed the norm of the visual state change $||f_v(v_{t+\delta}) - f_v(v_{t-\delta})||$ around MoIs and around randomly chosen timestamps. Tab. S2 confirms this claim. We also measured how well the `AStC` loss learns the association between the audio and the visual state change in the forward direction. Specifically, we calculated the average similarity $sim(\Delta v_t^{frwd}, a_t)$ within each of the three categories (T1, T2, T3). Tab. S2 shows a comparison of this forward association score between RepLAI and the AVID baseline. As expected, RepLAI learns better associations between the audio and visual state changes than AVID. More importantly, despite being are harder to identify, RepLAI still performs relatively well among T3 type interactions. This shows that, even for actions like washing and stirring, there are still slight visual state changes that the model can learn.

| Method | Mean Average Precision | | | Norm of Visual State Change | | | Average Similarity | | |
|--------|------|------|-------|------|------|-------|------|------|-------|
| | **T1** | **T2** | **T3** | **T1** | **T2** | **T3** | **T1** | **T2** | **T3** |
| AVID | 34.5 | 22.8 | 10.64 | 34.5 | 22.8 | 10.64 | 34.5 | 22.8 | 10.64 |
| RepLAI | 46.22 | 29.47 | 14.78 | 46.22 | 29.47 | 14.78 | 46.22 | 29.47 | 14.78 |

**Table S2:** Assessment of generalization ability of our method

| Method | visual state change ↑ |
|--------|------|
| Random location | 2.73 |
| Moment of Interaction (MoI) | 3.14 |

**Table S3:** Evaluating the detected moments of interaction

### B.2    Detection of moments of interaction

In the main paper, we showed that MoI detection improves representation quality. We evaluated the utility of moments of interaction (MoI) through their impact on representation quality and performance on multiple downstream tasks. Particularly, comparing rows (6) and (8) of Tables 1 and 2 of the main paper demonstrates that sampling training clips around MoIs improve representation quality and transfer.

We believe that MoI detection is especially useful for finding moments in the video with more perceptible visual state changes. We validate this by computing the norm of the difference between the before and after visual state for a detected MoI (averaged over all detected MoIs). A higher visual state change norm indicates that the model is able to detect locations in the video that have a significant and meaningful visual state change. From the Tab. S3, we observe that the norm of visual state change around the detected MoIs is significantly higher than that around randomly picked

locations. This validates that MoI is more effective in picking locations with relatively better visual state change. This, in turn, provides a richer signal to the model to learn better representations and provide stronger performance on downstream tasks.

## C   Qualitative Analysis

### C.1   Detected MoIs

In this section, we visualize of the moments of interaction detected with the help of spectrogram in several videos (Fig. S1, Fig. S2, and Fig. S3). While not perfect, we observe that sharp changes in the spectrogram energy correlate well with moments of interaction. Several of these moments are captured in Fig. S1, Fig. S2, and Fig. S3, such as opening drawers, putting down objects, cutting vegetables, etc.

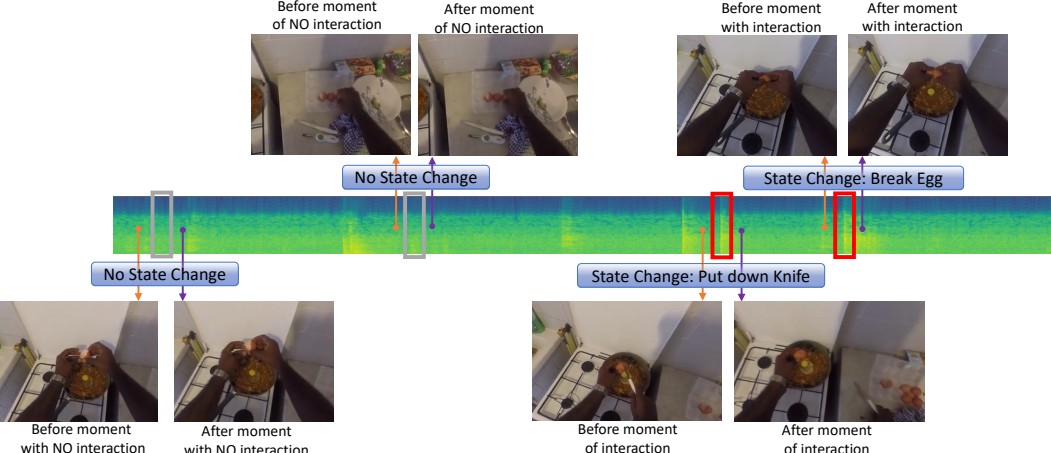

**Figure S1:** The above visualization shows the spectrogram of a video containing the action of putting down knife and breaking egg. The gray indicate the random moments with no moment of interaction and red indicate the moments of interaction.

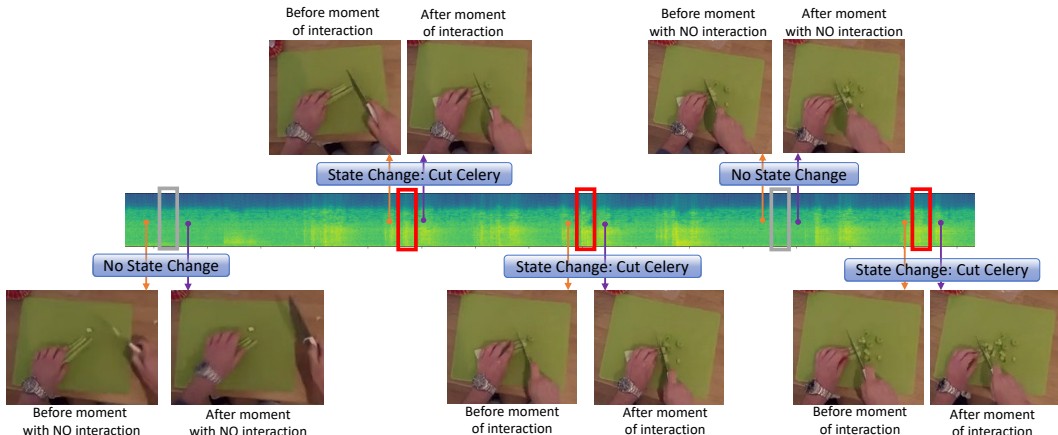

**Figure S2:** The above visualization shows the spectrogram of a video containing the action of cutting celery. The gray indicate the random moments with no moment of interaction and red indicate the moments of interaction.

### C.2   Audio-visual correspondence analysis

We analyse the audio-visual correspondence learned by our method, *RepLAI* and compare it with *AVID* [5] by generating a t-SNE plot in Fig. S4. This correspondence is helpful in assessing how well

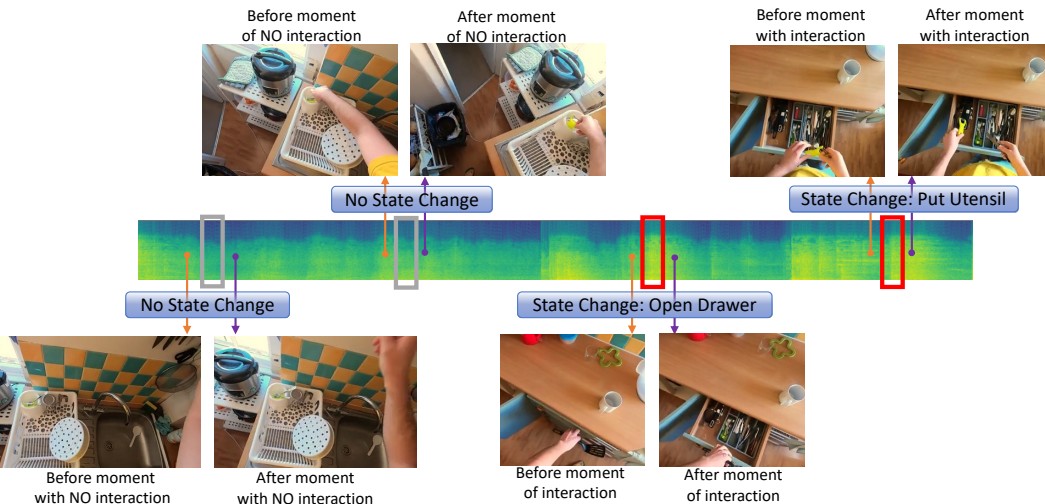

**Figure S3:** The above visualization shows the spectrogram of a video containing the action of opening drawer and putting cutlery. The gray indicate the random moments with no moment of interaction and red indicate the moments of interaction.

the model is able to predict if a short video clip and an audio clip correspond with each other or not. For a simpler visualization, a small dataset consisting of a single participant in EPIC-KITCHENS-100 is taken and split into clips of 0.5s. Both the audio and video features are visualized in the same space in the t-SNE plot and represented by gray dots. A few examples are selected randomly and their visual representation as well their audio representation is shown in colors. It can be observed that the audio-visual representation dots are closer in *RepLAI* representing better audio-visual correspondence compared to *AVID*. This indicates that the `AStC` is helpful in enhancing the correspondence learned between the video and audio.

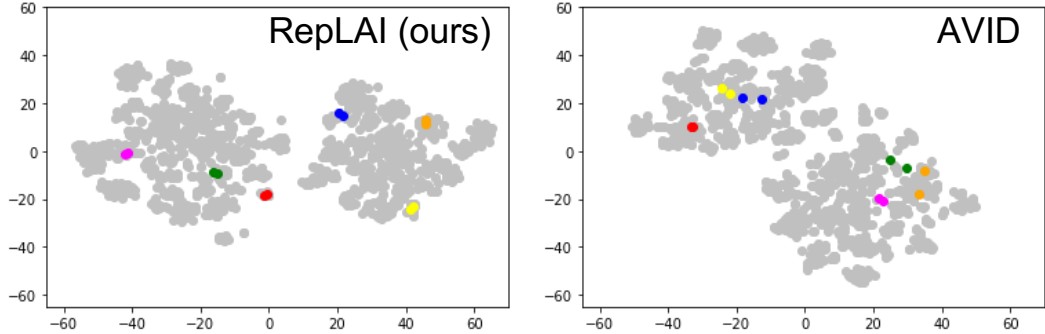

**Figure S4:** t-SNE visualization of the audio-visual feature representations learned by *RepLAI* and *AVID*. For a simpler visualization, we consider all the videos of a single participant. The gray dots represent both the audio and visual features in the same space. Randomly 6 examples are chosen and their two dots are shown in colors representing the visual features and the audio features. Closer the two dots are, better the audio visual correspondence.