# OpenReview forum: "Learning State-Aware Visual Representations from Audible Interactions"
_NeurIPS.cc/2022/Conference — NeurIPS 2022 Accept_

### Official Review · Reviewer_wUoy · 2022-07-09

**Rating:** 6
**Confidence:** 4
**Soundness:** 3 good
**Presentation:** 3 good
**Contribution:** 3 good

**Summary:**

This paper proposes a new way for self-supervised learning with egocentric videos to learn from audible interactions. Specifically, it uses audio signals to identify the state changes and use these transition states to learn the audio-visual correlations. The proposed method performs much better than a recent method AVID [42] and the ablation study shows the effectiveness of each model component.

**Questions:**

My main question is the generalization ability of the proposed method, i.e., increasing the probability of associating the audio with the visual state change in the forward direction but decreasing the probability in the backward direction. It can work well for activities like opening/closing the door intuitively. However, how can it work for others like stirring and washing?

**Limitations:**

The limitations and potential negative impact are discussed in the supplementary.

**Strengths And Weaknesses:**

Strengths

- The idea of using audio to identify the state changes for learning audio-visual correlation is novel and seems effective.
- The paper is well written and easy to read
- The performance of this method is good.

Weaknesses

- The example used for illustrating Eq. 1 are not convincing. The given example is the audio of closing the door should be (1) similar to the visual transition opened door → closed door and (2) dissimilar to the (backwards) transition closed → open. This activity has strong temporal order. But how to deal with activities like stirring and washing?
- There is lack of the fully-supervised performance for comparison in experiments.

---

> ### Author Response · Authors · 2022-08-02
> **Thanks and Response to Reviewer wUoy - Part A**
>
> We thank the reviewer for their valuable feedback and suggestions.
>
> > My main question is the generalization ability of the proposed method...It can work well for activities like opening/closing the door intuitively. However, how can it work for others like stirring and washing?
>
> To provide further insights onto the generalization ability of the proposed method, we conduct an experiment to assess how discriminative the learned representations are for different types of activities. For this experiment, we first categorize the activities based on the nature of the transition:
>
> T1) “irreversible” audible interactions with backward transition never happening (e.g., cutting vegetables);
> T2) “reversible” audible interactions with backward transition likely to happen  (e.g., open/close fridge);
> T3) audible interaction with no transition direction (e.g., stirring).
>
> Clearly, AStC can learn from both T1 and T2 interactions, as they are associated with visual state changes. Although T1 interactions are never seen in reverse order, the model still benefits from knowing what is the correct order, as this leads to more state-aware representations. As for T3 type interactions, they can indeed be a failure mode of the AStC objective, if they cause NO change in the visual state of the environment.
>
> However, we highlight that ALL self-supervised objectives have failure modes. For example, the AVC objective cannot learn to associate audio and video if the audio is silent. Even standard image-based contrastive frameworks like MoCo or SimCLR cannot learn when the sampled crops do not capture parts of the same object. In fact, self-supervised pre-training is not meant to be perfect. It is only meant to provide a proxy for learning representations, that can be tuned on a downstream task. Despite not being able to learn from T3 type interactions, AStC significantly improves the learned representations when considered in conjunction with AVC (for example, see the gains between rows 4 and 7 of Tables 1 and 2).
>
> To provide further insights, we computed the mean average precision for each type of interaction, after training for the downstream action recognition task. We also compared this metric using both the finetuned AVID baseline and RepLAI models. The results, shown in the table below, indicates the RepLAI performs significantly better than the baseline across all categories of transition/direction, showing that RepLAI (which includes both AVC and AStC) is generic enough to enhance representations for all types of interactions.
>
> Mean average precision score:
> | Method               |   T1      |   T2      |   T3      |
> |------------------    |:-----:    |:-----:    |:-----:    |
> | AVID (Finetuned)     | 34.50     | 22.80     | 10.64     |
> | RepLAI (Ours)        | **46.22**     | **29.47**     | **14.78**     |
>
> Furthermore, we observed that MoI detection helps finding timestamps that have more perceptible visual state change (even for T3 type interactions). To see this, we computed the norm of the visual state change $\|\|f_v(v_{t-\delta})-f_v(v_{t-\delta})\|\|_2$ around MoIs and around randomly chosen timestamps. The results, shown below, confirm this claim.
>
> Norm of visual state change:
> | Method                          |  T1      |  T2      |  T3      |
> |-----------------------------    |:----:    |:----:    |:----:    |
> | Random location                 | 2.77     | 2.76     | 2.61     |
> | Moment of Interaction (MoI)     | **3.19**     | **3.17**     | **3.01**     |
>
>
> Finally, we also measured how well the AStC loss learns the association between the audio and the visual state change in the forward direction. Specifically, we calculated the average similarity $ sim ( \Delta v_t^{frwd} , a_t)$ within each of the three categories (T1, T2, T3). The table below shows a comparison of this forward association score between our approach (RepLAI) and the AVID baseline. As expected, RepLAI learns better associations between the audio and visual state changes than AVID. More importantly, despite being are harder to identify, RepLAI still performs relatively well among T3 type interactions. This shows that, even for actions like washing and stirring, there are still slight visual state changes that the model can learn.
>
> Average similarity:
> | Method            |   T1      |   T2      |   T3      |
> |---------------    |:-----:    |:-----:    |:-----:    |
> | AVID              | 0.364     | 0.357     | 0.346     |
> | RepLAI (Ours)     | **0.782**     | **0.764**     | **0.646**     |
>
> > There is lack of fully-supervised performance for comparison in experiments.
>
> We already provide some fully supervised results on Ego4D (Table 2, rows S1, S2, S3) and have a discussion on that in Lines 325-329. We acknowledge reviewer’s suggestion and will provide additional supervised results in Table 1, using the same R(2+1)D-18 architecture pre-trained on Kinetics-400. Unfortunately, we were not able to finish this result during the rebuttal period.

---

> > ### Comment · Reviewer_wUoy · 2022-08-10
> > **Thanks!**
> >
> > Thank you for the response.
> >
> > I agree with the other two reviewers that the comparison with other state-of-the-art methods is not sufficient in the original submission and the additional comparisons provided should be added into the paper. I agree with Reviewer coBj that some examples should also be provided in the paper for understanding the method.
> >
> > I agree with Reviewer WY3G that the generalization ability of the method is not convincing, the authors' response has partly addressed my concern, but I think it is acceptable that this is left for future works. And I agree with Reviewer coBj that the idea of using audio to identify the state changes for learning audio-visual correlation is interesting. I incline to keep my acceptance rating and hope the source code can be released for future works.

---

> ### Author Response · Authors · 2022-08-08
> **Requesting feedback**
>
> From the review, we believe that the main concern raised is the generalization ability of the proposed method (for example, how to deal with actions _without strong temporal order_ like stirring). We addressed this concern by conducting a thorough analysis of the impact of different types of interactions on the learned representations. We add three new analysis and corresponding insights, particularly:
>
> (A). **Mean average precision among three types of interactions:** This shows that our model improves over baselines, regardless of interaction type.
>
> (B). **Norm of the visual state change around an MoI**: Regardless of the interaction type, we show bigger changes in visual states occur at the detected MoIs than at uniformly random timestamps. This shows that even for interactions without strong temporal order, our model can focus on visual cues that do change over time.
>
> (C). **Association score between the audio and the visual state change:** This shows that AStC is able to associate (better than prior work) the audio with the visual state change representation, even for interactions without strong temporal order.
>
> We believe we concretely address the main concerns shared in the review. Is our response (and planned changes to the manuscript) satisfactory? If not, please let us know how we can better address these concerns.
>
> Thank you so much for sharing constructive feedback to help improve our manuscript!

---

### Official Review · Reviewer_WY3G · 2022-07-11

**Rating:** 4
**Confidence:** 4
**Soundness:** 2 fair
**Presentation:** 3 good
**Contribution:** 2 fair

**Summary:**

This paper proposes leveraging audible state changes for self-supervised representation learning from long-form egocentric videos. Authors initially detect the timestamps where interesting interactions occur (MoI) and then compute a cross modal contrastive objective where the natural transition (before, interaction, after) is more likely than one which is temporally reverse (after, interaction, before).

**Questions:**

- How do you handle the audio diversity of semantically similar interactions? For example, “cutting cucumber” on a wooden board will sound different from doing it on a plastic board, or sound of “sautéing” mushrooms/onions in a pan is meaningfully influenced by the oven/pan/oil temperature.

- As part of ablation studies, have you tried a linear head for $h_{\Delta V}^{AStC}$ since forward delta seems to be the negative of the backward one.

- AVC loss shown in Figure 3.b encourages the embedding of $V_{t-\delta}$ to be close to $V_{t+\delta}$ by both anchoring on audio embedding at t while the AStC encourages them to be different. These two objective functions, to the best of my understanding are pulling in opposite directions! Would not it make more sense to choose the audio at $t-\delta$ and $t+\delta$, instead of t, when computing two AVC losses?

- In Table 1, Top 5 Acc: AVC seems to be doing most of the job while w/o AStC and MoI, performance on “verb” is almost maintained (~73%) but the pattern is different for “noun”. Any insights?


**Limitations:**

Authors have addressed the limitations

**Strengths And Weaknesses:**

Paper is clear in presentation and has provided an interesting view to self-supervised multi-modal representation learning in egocentric videos with audible state change. The idea is pretty close to "Actions ∼ Transformations" <https://arxiv.org/pdf/1512.00795.pdf> which unfortunately is missing from the references. Below are my detailed comments.

- Line 48-49: Authors argue that AVC objective leads to representations that are not informative of the changes over time. If indeed the sound of “opening fridge” is different from that of “closing fridge”, AVC will encourage video representation of “opening fridge” to be more similar to the audio representation of “opening fridge” than “closing fridge” since audio embedding of the latter will be served as a negative instance within the contrastive framework. Note that, you do not have to sample after the fact e.g once the fridge is closed/opened, instead you can sample during the action e.g while fridge is being opened/closed, something that working with video in a cross-modal contrastive setup easily allows you to do. With that said, I would like to see authors clarifying their point on the lack of AVC’s suitability for the task specially given the ablation studies that show dropping AVC vs AStC is not that much different.

- On finding MoI: As paper mentions, egocentric videos are naturally long-form, as they continuously capture daily activities. Hence, a person moving in an environment i.e change in location, even without any interaction can result in changes in audio spectrogram. For example, a person is cooking some food in the kitchen, then walks to the living room to pick up a book. The fact that there is a fan or stereo playing in the living room, will result in a change in audio (note that the sound of fan or stereo is not necessarily audible in the kitchen), however those are not MoI since there has been no interaction with either stereo or fan yet from Sec 3.2 it seems to me that the proposed MoI approach should pick those. In nutshell, change in audio is not only as a result of human-object interaction, it can be due to change of location or variant environment as well and I cannot see how the proposed MoI detection method can work in a realistic environment.

- I am not convinced that learning from audible state change as described in Sec 3.3 is generic enough. For example, the visual state is different after hearing the sound of MoI for “opening fridge” , while before state shows a closed fridge, the after state depicts an open one. Also, due to “distinct” sound of “opening fridge” the backward transition should be less likely. Now consider the example of “cutting cucumber”, the backward transition almost never happens (humans usually don’t stitch cucumber slices together!), while it is reasonable to go from a closed fridge to an open one and vice versa. There are also cases which despite audible MoI, visuals look almost identical in before and after like “stirring a pot”. I suspect that more clear performance gains seen on Ego4D versus Epic-Kitchens is partly related to the state change properties which are more prominent in the former dataset. I would like to hear authors feedback on the different aforementioned types of interactions and why their proposed model should work in a self-supervised setup where we don’t know which type of these interactions are included in the training data.

- Line 269: I do not think it is fair to compare rows 2 and 7 since row 2 has been only trained on AudioSet which is very different from evaluation egocentric datasets. To see the true additional value of your contributions (use of MoI and AStC loss), AVID should be fine-tuned on the egocentric datasets (equally RepLAI w/o AStC and MoI)

---

> ### Author Response · Authors · 2022-08-02
> **Thanks and Response to Reviewer WY3G - Part A**
>
> We thank the reviewer for their valuable feedback and suggestions.
>
> > Authors argue that the AVC objective leads to representations that are not informative of the changes over time…Can the authors clarify their point on the lack of AVC’s suitability for the task specially given the ablation studies that show dropping AVC vs AStC is not that much different.
>
> While AVC helps to learn effective correspondence between the visual states and their associated audio, it is invariant to the visual state change by design. So, although AVC can learn a discriminative representation for “opening fridge” (which is indeed different from “closing fridge”), it provides NO incentive for the model to learn the higher-order information of temporal change in the visual state. This information is necessary for the model to build a temporal understanding of whether we are at the beginning of the action (e.g., holding the handle to open the fridge) or at the end of the action (e.g., opening the fridge to grab something). As mentioned in line 297-305, our newly introduced AStC loss complements and mitigates this shortcoming of AVC by operating directly on the direction of the visual state change.
>
> AStC is designed to make the model learn this temporal change in visual state by correctly associating the forward direction of change ($\Delta v_t^{\text{frwd}}$) with the audio (Equation 4) and reducing the association of the backward direction of change ($\Delta v_t^{\text{bkwd}}$) with the audio (Equation 5). By directly working on the direction of visual state change ($\Delta x$) rather than on the visual states ($x$), AStc allows the model to be more aware about the transitions in visual states over the course of any action or interaction in the environment.
>
> We would like to clarify that we do not suggest the equality of the AVC and AStC terms or the superiority of one over the other. By _complementing_ each other, AVC and AStc when combined together (Row 7 of Table 1 and Table 2 of main paper), RepLAI achieves the best performance across all downstream tasks compared to other settings of not having either AVC or AStc (Row 3, 4 of Table 1 and Table 2 of main paper)
>
> > …given the ablation studies that show dropping AVC vs AStC is not that much different.
>
> The performances shown, for example, in Rows 3 and 4 of Tab. 1 are indeed similar (29.9% vs 29.3% top1 verb accuracy and 10.5% vs 9.7% top1 noun accuracy), but this is likely just a coincidence. We highlight that AVC and AStC are required to learn very different representations by design - AVC focuses on the direct audio-visual correspondences, and AStC focuses on the direction of visual change. Thus, the fact that AStC alone achieves similar performance to the more common AVC objective in both Tables 1 and 2 already shows the potential of AStC.
>
> The second important insight is that AVC and AStC are indeed complementary to each other. Their combination (Row 7 of Table 1 and Table 2 of paper) achieves the best performance across all downstream tasks compared to other settings of not having either AVC or AStc. In the absence of either (Row 3 and Row 4), the model learns sub-optimal representations and achieves lower performance. For example, in Tab. 1, the combination yields (**31.7, 11.2**) top1 accuracy for (verbs, nouns), compared to (29.9, 10.5) without AVC. Similar gains are found in Tab. 2 for the Ego4D dataset on different downstream tasks.

---

> > ### Author Response · Authors · 2022-08-02
> > **Thanks and Response to Reviewer WY3G - Part B**
> >
> > > In nutshell, change in audio is not only as a result of human-object interaction, it can be due to change of location or variant environment as well.
> >
> > In Lines 53-54 and Lines 135-137, we intend to convey that an MoI can be any form of action or interaction among the entities in the environment that leads to a perceptible visual state change. Thus, MoIs are not restricted to just human-object and object-object interactions and can be associated with any form of interaction within the environment. We hypothesize that whenever there is such a visual state change, there is a high probability for this change to be accompanied with a distinct/characteristic audio pattern (Lines 53-54) leading to an MoI detection. Such a formulation of MoI enables our method to focus on timestamps in the video that provide relatively richer audio-visual signals for the model to learn meaningful feature representations that understand both visual states and temporal change in visual states. Note that we purposefully intend to detect generic enough MoIs to accommodate the unconstrained and unlabeled nature of the training data, and prevent being restricted to a closed set of human interactions.
> >
> > We agree with the reviewer that there will be a change in the audio in the reviewer’s example when a person walks to the living room and there is a fan or stereo playing. As a result, MoI detection will indeed pick this timestamp. But given the above mentioned definition, we posit that this change in audio is a valid MoI, as it is associated with the person changing its location (an outcome of the interaction of the person with their environment). This will be accompanied by a significant change in the visual state of the environment along with a distinct audio pattern. We will paraphrase Section 3.2 to elucidate this definition of Moment of Interaction (MoI).
> >
> > > I cannot see how MoI detection works in a realistic environment.
> >
> > Both Ego4D[1] and Epic-Kitchens[2] are challenging, realistic environments and have portions of video that do not have clean audio. Ego4D consists of several daily activities in the home, workplace, social setting, leisure and commuting which are unscripted and ``in the wild'' that represent natural interactions (Page 2 of Ego4D [1]). Epic-Kitchens also includes long-term, unscripted videos of human-object interactions with everyday objects in a kitchen (Page 2 of Epic-Kitchens [2]). The improved performance over multiple downstream tasks~(Table 1, 2) achieved by detecting MoIs indicates that the proposed approach is generic enough to work in a realistic environment.
> >
> > [1]. Grauman, Kristen, et al. "Ego4d: Around the world in 3,000 hours of egocentric video." Proceedings of the IEEE/CVF Conference on Computer Vision and Pattern Recognition. 2022.
> >
> > [2]. Damen, Dima, et al. "Rescaling egocentric vision." arXiv preprint arXiv:2006.13256 (2020).

---

> > > ### Author Response · Authors · 2022-08-02
> > > **Thanks and Response to Reviewer WY3G - Part C**
> > >
> > > > I am not convinced that learning from audible state change is generic enough. I would like to hear authors' feedback on different types of interactions and why their proposed model should work in a self-supervised setup where we don’t know which type of these interactions are included in the training data.
> > >
> > > To provide further insights on how AVC and AStC learn from different types of interactions, we categorize all actions in Epic-Kitchens into three types:
> > > T1) “irreversible” audible interactions with backward transition never happening (e.g., cutting vegetables);
> > > T2) “reversible” audible interactions with backward transition likely to happen  (e.g., open/close fridge);
> > > T3) audible interaction with no transition direction (e.g., stirring).
> > >
> > > AStC naturally learns from both T1 and T2 interactions, as they are associated with a visual state change (the reviewer seems to suggest that AStC can handle T2 but not T1). Although T1 interactions are never seen in reverse order, the model still benefits from knowing what is the correct order, as this leads to more state-aware representations. As for T3 type interactions, they can indeed be a failure mode of the AStC objective, if they cause __no change__ in the visual state of the environment.
> > >
> > > However, we highlight that ALL self-supervised objectives have failure modes. For example, the AVC objective cannot learn to associate audio and video if the audio is silent. Even standard image-based contrastive frameworks like MoCo or SimCLR cannot learn when the sampled crops do not capture parts of the same object. In fact, self-supervised pre-training is not meant to be __perfect__. It is simply meant to provide a pretext task for learning representations, that can be adapted (via sample-efficient tuning) on a downstream task. Despite not being able to learn from T3 type interactions, AStC significantly improves the learned representations when considered in conjunction with AVC (for example, see the gains between rows 4 and 7 of Tables 1 and 2).
> > >
> > > To provide further insights, we take the mean average precision score for each type of interaction, after training for the downstream action recognition task. We also compared this metric using both the finetuned AVID baseline and RepLAI models. The results, shown in the table below, indicates the RepLAI performs significantly better than the baseline across all categories of transition/direction, showing that RepLAI (which includes both AVC and AStC) is generic enough to enhance representations for all types of interactions.
> > >
> > > Mean average precision score for three types of interaction:
> > >
> > > | Method               |   T1      |   T2      |   T3      |
> > > |------------------    |:-----:    |:-----:    |:-----:    |
> > > | AVID     | 34.50     | 22.80     | 10.64     |
> > > | RepLAI (Ours)        | **46.22**     | **29.47**     | **14.78**     |
> > >
> > > Furthermore, we observed that MoI detection helps finding timestamps that have more perceptible visual state change (even for T3 type interactions). To see this, we computed the norm of the visual state change $\|\|f_v(v_{t-\delta})-f_v(v_{t-\delta})\|\|_2$ around MoIs and around randomly chosen timestamps. The results, shown below, confirm this claim.
> > >
> > > Norm of visual state change for three types of interaction:
> > >
> > > | Method                          |  T1      |  T2      |  T3      |
> > > |-----------------------------    |:----:    |:----:    |:----:    |
> > > | Random location                 | 2.77     | 2.76     | 2.61     |
> > > | Moment of Interaction (MoI)     | **3.19**     | **3.17**     | **3.01**     |
> > >
> > > > It is not fair to compare rows 2 and 7 since row 2 has been only trained on AudioSet which is very different from evaluation egocentric datasets. AVID should be fine-tuned on the egocentric datasets.
> > >
> > > We acknowledge reviewer's suggestion and finetune AVID on Epic-Kitchens using only AVC (i.e., RepLAI w/o AStc and MoI). The results for this setup are tabulated in the table below, alongside other settings from Table 1. We observe that finetuning AVID on the egocentric data does improve performance, but it is still significantly behind the proposed RepLAI method. We will add these results to the paper, as well as the equivalent results on Ego4D~(which we couldn't finish due to time constraints).
> > >
> > > |                      | Top-1 Acc     |           | Top-5 Acc     |           |
> > > |------------------    |:---------:    |:-----:    |:---------:    |:-----:    |
> > > | Method               |    Verb       |  Noun     |    Verb       |  Noun     |
> > > | AVID                 |   26.62       |  9.00     |   69.79       | 25.50     |
> > > | AVID (Finetuned)     |   27.25       |  9.10     |   69.94       | 26.14     |
> > > | RepLAI w/o AStC      |   29.29       |  9.67     |   73.33       | 29.54     |
> > > | RepLAI w/o MoI       |   28.71       |  8.33     |   73.17       | 27.29     |
> > > | RepLAI (Ours)        |   **31.71**       | **11.25**     |   **73.54**       | **30.54**     |

---

> > > > ### Author Response · Authors · 2022-08-02
> > > > **Thanks and Response to Reviewer WY3G - Part D**
> > > >
> > > > > How do you handle the audio diversity of semantically similar interactions? For example, “cutting cucumber” on a wooden board will sound different from doing it on a plastic board, or sound of “sautéing” mushrooms/onions in a pan is meaningfully influenced by the oven/pan/oil temperature.
> > > >
> > > > From an audible state-change (AStC) perspective, we expect that the visual representations of cutting vegetables to be similar i.e. invariant to the type of cutting board. This is because the only visible change is ‘whole vegetable’ to ‘cut vegetable’ and the cutting board is unaffected by this visual state change.
> > > > However, from an audio-visual correspondence (AVC) perspective, we expect that the different sounds produced by the wooden vs plastic boards would allow the model to develop visual features that are discriminative of the type of cutting board.
> > > > However, since AVC and AStC operate on different projections of the audio and visual space, these two goals do not contradict each other.
> > > >
> > > > > As part of ablation studies, have you tried a linear head for $h_{\Delta V}^{AStC}$ since forward delta seems to be the negative of the backward one.
> > > >
> > > > We acknowledge reviewer's suggestion and train RepLAI with a linear head for the visual state change projection. The results are shown in the table below where we compare RepLAI trained with linear head and RepLAI trained with non-linear head (ours). From the table, we can observe that although the difference between the two settings is small, the non-linear head design performs slightly better.
> > > >
> > > > |                           | Top-1 Acc     |           | Top-5 Acc     |           |
> > > > |-----------------------    |-----------    |-------    |-----------    |-------    |
> > > > | Method                    | Verb          | Noun      | Verb          | Noun      |
> > > > | RepLAI w/ linear head     | 31.02         | 10.85     | 73.12         | 30.08     |
> > > > | RepLAI (Ours)             | **31.71**         | **11.25**     | **73.54**         | **30.54**     |
> > > >
> > > > > AVC loss shown in Figure 3.b encourages the embedding of $v_{t-\delta}$ to be close to $v_{t+\delta}$ by both anchoring on audio embedding at t while the AStC encourages them to be different. These two objective functions, to the best of my understanding are pulling in opposite directions!
> > > >
> > > > This is a misunderstanding. AVC and AStC do NOT contradict each other, as they operate on different feature spaces. As we showed in Fig 3, the representations that are matched in the AVC and AStC objectives are obtained from *different* projections of the underlying representations $f_V(v)$ and $f_A(a)$, ie, using different projection heads $h^{AVC}$ and $h^{AStC}$. Note that the different projection heads avoid this contradiction, as the model can dedicate part of the $f_V(v)$ representation to be invariant to state-changes (and thus good for AVC), and another part to be discriminative of state changes (to satisfy AStC).
> > > >
> > > > > Would not it make more sense to choose the audio at $t-\delta$ and $t+\delta$, instead of $t$, when computing two AVC losses?
> > > >
> > > > It wouldn’t make much difference as audio clips are relatively large (compared to $\delta$), and so both audio clips would overlap significantly. Having said that, this implementation does make perfect sense, and we did consider it. However, given that using a single audio clip at time t does NOT impose any limitation (as described in the answer to the previous comment), we decided to go with this implementation to reduce computation (ie., to reduce the number of forward and backward passes required at each iteration).
> > > >
> > > > > In Table 1, Top 5 Acc: AVC seems to be doing most of the job while w/o AStC and MoI, performance on “verb” is almost maintained (~73%) but the pattern is different for “noun”. Any insights?
> > > >
> > > > It is difficult to make a concrete remark here, especially because of the broad nature of the top-5 metric as well as the head trained after self-supervised pre-training. It is likely that although having five predictions reduces the difficulty of classifying certain competing/similar verbs, it does not help in distinguishing similar noun classes.

---

> > > > > ### Author Response · Authors · 2022-08-02
> > > > > **Thanks and Response to Reviewer WY3G - Part E**
> > > > >
> > > > > > The idea is pretty close to "Actions ∼ Transformations" which unfortunately is missing from the references.
> > > > >
> > > > > We thank the reviewer for bringing this paper to our attention. We will duly cite this paper as they learn from the same concept of visual state change. Although the underlying concept -- transformation or change in the environment caused by an action -- is similar to ours, the settings, the algorithm and the overall contributions are quite different from our work.
> > > > >
> > > > > - We focus on the *multi-modal setting*, which has two advantages: 1) state changes are often indicated by audio and can be therefore better localized and 2) the audio itself is often discriminative of what type of state change took place.
> > > > >
> > > > > - Unlike ”Actions~Transformations”, we focus on the *self-supervised* representation learning problem. As a result, we cannot leverage class information to learn a class specific transformation between the before and after states.
> > > > >
> > > > > -  “Actions ∼ Transformations” takes the simplifying assumption of working with trimmed videos, while we consider the more challenging (and realistic) setting containing arbitrarily long, untrimmed videos of daily activities.

---

> ### Author Response · Authors · 2022-08-08
> **Requesting feedback**
>
> From the review, we believe that there were four main concerns preventing the reviewer from providing a higher score. In our rebuttal, we directly address each of these concerns. Is our response (and planned changes to the manuscript) satisfactory? If not, please let us know how we can better address these concerns.
>
> **Concern #1:** AStC may not be necessary since AVC can already learn representations aware of changes over time if associated with different sounds (eg different representations for opening and closing a fridge)
>
> **Response:** To address this concern, we pointed out that, unlike AStC, AVC provides _no incentive_ for the model to learn the temporal change in the visual state. This information allows the model to learn whether a clip is from the beginning or the end of the action, and thus encourages the model to learn more state aware representations. The complementarity between AVC and AStC is also experimentally validated throughout our experiments.
>
> **Concern #2:** Learning from MoIs may not be appropriate, as audio changes can be caused by more than human-object interactions
>
> **Response:** To address this concern, we clarified that MoI should be thought of as _any form of interaction_ that leads to a perceptible visual state change (Lines 53-54, 135-137), since the AStC loss can learn from any such interactions. _MoIs are not restricted to just human-object interactions_, and can be associated with any form of interaction within the environment. We also experimentally showed the benefits of MoI detection for several downstream tasks on realistic datasets like EPIC-Kitchens and Ego4D.
>
> **Concern #3:** AStC may not be general enough, since it may not work with interactions that are irreversible (like cutting vegetables) or interactions that cause no visual changes (like stirring a pot)
>
> **Response:** To address this, we add a new analysis on the impact of our method on the representations for three types of interactions. We measured how our discriminative features (using mean average precision), show improved performance compared regardless of action type. Using metrics like the norm of visual state changes, we show that even for an interaction like stirring a pot, the model can find visual cues that do change over time, indicating that this potential failure mode of AStC is more rare than one might expect.
>
> **Concern #4:** Fine-tune AVID baseline on egocentric datasets for a fair comparison
>
> **Response:** We include this additional run in our response. The same trends hold and this baseline helps strengthen the results.
>
> We thank the reviewer again for their time and effort towards improving the manuscript.

---

### Official Review · Reviewer_coBj · 2022-07-11

**Rating:** 6
**Confidence:** 3
**Soundness:** 3 good
**Presentation:** 3 good
**Contribution:** 3 good

**Summary:**

In this paper authors introduce a self-supervised learning method from audio-visual data. Specifically, the method revolves around using moments of interaction to learning meaningful audio-visual relation. Authors split the learning in two losses: an standard audio-visual loss, and a loss around the audible change of state and its relation to the visual change of state. Authors evaluate their proposed method in EpicKitchens and Ego4D.


**Questions:**

My concerns are listed in Weaknesses. My main questions to the authors would be:

- Have the authors considered adding additional baselines when comparing to the state-of-the-art? If not, why not?

- Which evidence do the authors have that the detection of change works properly if they don't have any evaluation for that signal.

- In general, could the authors show some more examples of moments of interaction (similar to the ones in supplementary) in the main paper. I think it would be specially interesting to see examples where the detection of those moments works and other where it fails.

**Limitations:**

Authors discuss the limitations of the method in 4.2. I would suggest expanding that a bit with general negative societal impact that working with ego video can have. I believe egocentric video can be used to track everyday's life which could have some negative impact.

**Strengths And Weaknesses:**

**Strengths**:

- S1. Audio-visual self-supervised learning is a powerful tool to learn representations. However, typical approaches do not exploit the semantic importance of the moments where to sample the audio. This work is a step towards using the audio content in a most meaningful way.

- S2. The results show how the method improves over the baseline with standard audio-visual learning.

- S3. Authors evaluate in two very relevant well-known ego centric benchmarks. I believe that is important as it makes the paper stronger.

- S4. Ego-centric data is going to be becoming more important in the near future. With the progress of recording devices and embodied research, works along the lines of this paper will become more relevant.

- S5. I specially like the idea of detecting state changes through audio. According to the paper the procedure is quite simple and the self-supervised method certainly benefits from seeing samples around that moment.

**Weaknesses**

- W1. Authors compare with a single baselines for audio-visual learning. I think other works such as MMV (Alayrac, NeuriPS 2020),  Evolving Losses (Piergiovanni, CVPR 2020), Brave (Recasens, ICCV 2021), XDC (Alwassel, NeurIPS 2020) are very relevant in the community. I understand the topic is slightly different and authors cannot retrain with all the baselines, but I still believe that having a single baseline such as AVID is insufficient for a publication.

- W2. The authors are missing citations for some of the papers mentioned in W1 (e.g. MMV, Evolving Losses). I think those works are important in the space of self-supervised audio-visual learning.

- W3. I think authors do not properly evaluate the ability of their model to detect moments of interaction. The description of the method is very complete and in Table 1 they ablate using the method, but I think it would be good to somehow evaluate whether the proposed method works.

- W4. I am missing some examples of moments of interaction I understand the reasoning behind the methodology of using the reverse clip as negative, but I believe the readers would benefit from a few visual examples to understand that.

---

> ### Author Response · Authors · 2022-08-02
> **Thanks and Response to Reviewer coBj - Part A**
>
> We thank the reviewer for their valuable feedback and suggestions.
>
> > Have the authors considered adding additional baselines when comparing to the state-of-the-art? If not, why not?
>
> We thank the reviewer for the constructive recommendation. We will incorporate the suggestion in the final revision. To do so, we used XDC pre-trained on Audioset (Alwassel et al. 2020) as the baseline. Notably, unlike most other audio-visual self-supervised learning (SSL) methods, XDC’s self-supervised objective is based on deep clustering (most others are contrastive, akin to AVID). Below, we report results on the Epic-Kitchens. Rows (1) and (2) are exactly from the paper. Row (3) shows the performance of the pre-trained XDC model on the action recognition task, and row (4) shows the performance of our RepLAI model using XDC as the initialization (instead of AVID).
>
> |                                        | Top-1 Acc     |           | Top-5 Acc     |           |
> |------------------------------------    |:---------:    |:-----:    |:---------:    |:-----:    |
> | Method                                 |    Verb       |  Noun     |    Verb       |  Noun     |
> | (1). AVID (Morgado et al. 2021)        |   26.62       |  9.00     |   69.79       | 25.50     |
> | (2). RepLAI w/ AVID initialization     |   **31.71**       | **11.25**     |   **73.54**       | **30.54**     |
> | (3). XDC (Alwassel et al. 2020)        |   24.46       |  6.75     |   68.04       | 22.71     |
> | (4). RepLAI w/ XDC initialization      |   29.58       |  9.62     |   71.87       | 28.05     |
>
> XDC representations transfer slightly worse to Epic-Kitchen than AVID (row 1 vs 3). Our method performs better than XDC by significant margins (row 2 vs 3). RepLAI still makes strong improvements when using XDC as the initialization (row 3 vs 4). These results provide further evidence of the benefits of the proposed approach. In the final revision, we plan to complete the analysis with XDC initialization. We will include results on the Ego4D dataset and tasks.
>
> > The authors do not properly evaluate the ability to detect moments of interaction. What evidence do the authors have that the detection of change works properly?
>
> As suggested by the reviewer, we try to evaluate the ability of the model to detect moments of interaction. In our work, we show that MoI helps to improve performance by detecting locations in the video that have better perceptible visual state change. We validate this by computing the norm of the difference between the before and after visual state for a detected MoI (averaged over all detected MoIs). A higher visual state change norm indicates that MoI is able to detect locations in the video that have a significant and meaningful visual state change. We compare this with that computed over all randomly chosen locations in the training videos and tabulate the results below,
>
> | Method                          | $\|\|$ visual state change $\|\|$ $\uparrow$     |
> |-----------------------------    |:----:    |
> | Random location                 | 2.73     |
> | Moment of Interaction (MoI)     | **3.14**     |
>
> From the above table, we can observe that the magnitude of visual state change around detected MoI is significantly larger than that around randomly picked locations. This validates that MoI is more effective in picking locations with relatively better visual state change. This, in turn, provides a richer signal to the model to learn better representations and provide stronger performance on downstream tasks. Even if the detected moments of interest are less accurate, biasing the sampling of training clips towards moments in time is more conducive for learning representations, i.e., when interactions or actions are happening.
>
> Additionally, in our work, we indeed evaluate the utility of moments of interaction (MoI) through their impact on representation quality and performance on multiple downstream tasks. Particularly, comparing rows 5 and 7 of Tables 1 and 2 of the main paper demonstrates that sampling training clips based on our MoIs improve representation quality and transfer. We would also like to highlight that the proposed self-supervised objectives (Sec. 3.3 and 3.4) utilized to train the model are meaningful regardless of MoIs – comparing rows (2) and (5) in Tab. 1 of the main paper.

---

> > ### Author Response · Authors · 2022-08-02
> > **Thanks and Response to Reviewer coBj - Part B**
> >
> > > W2. The authors are missing citations for some of the papers mentioned in W1 (e.g. MMV, Evolving Losses).
> >
> > Thank you for pointing out the missing references. We will add them to the revised draft. We agree - MMV and Evolving Losses are important works in the audio-visual self-supervised learning (SSL) literature.
> >
> > > W4. Could the authors show more examples of moments of interaction? It would be specially interesting to see where the detection of those moments works and where it fails...I understand the reasoning behind the methodology of using the reverse clip as negative, but I believe the readers would benefit from a few visual examples to understand that.
> >
> > Due to space restrictions, we could only visualize three moments of interaction in the main paper (Fig. 1 and Fig. 4). More examples were included in Supplementary Material (Fig 1-3) which we will definitely expand them in a revised version, including qualitative visualization of reverse clips as negatives. Thank you for this constructive feedback.
> > An intuitive failure mode of a moment of interaction is when an action is not associated with a distinctive audio pattern, such as “look”. We’ll revise and incorporate this.

---

> > > ### Comment · Reviewer_coBj · 2022-08-09
> > > **Thanks!**
> > >
> > > Dear authors,
> > >
> > > Thank you for your detailed and informative reply. Most of my concerns are now resolved and I believe the paper should be accepted in the conference.
> > >
> > > Best wishes.

---

> > > > ### Author Response · Authors · 2022-08-09
> > > > **Thanks!**
> > > >
> > > > Thanks a lot for your efforts, sharing constructive feedback, and post-rebuttal response.

---

> ### Author Response · Authors · 2022-08-08
> **Requesting feedback**
>
> From the review, we believe that the main concerns were to:
>
> (1) consider baselines beyond AVID
>
> (2) evaluate the ability to detect moments of interaction (MoI)
>
> We addressed (1) by providing new results on EPIC-Kitchens for the suggested XDC baseline, as well as, RepLAI with an XDC initialization. We will include additional results on the Ego4D dataset with more time to revise.
>
> We addressed (2) by presenting a new analysis that measures the norm of the changes in visual representation. We show bigger changes in visual states occur at MoIs that we detect as compared to uniformly random timestamps. We point to further results in the main paper that show the impact of MoI on representation quality and downstream performance across multiple tasks.
>
> We believe that we concretely address the main concerns shared in the review. Is our response (and planned changes to the manuscript) satisfactory? If not, please let us know how we can better address these concerns.
>
> Thank you so much for your efforts to help improve our work!

---

### Author Response · Authors · 2022-08-02
**General Response for all Reviewers**

We thank all the reviewers for their valuable feedback and suggestions. In this paper, we introduce an audio-driven self-supervised method to learn representations of egocentric video of daily activities. Our approach uses audio in two novel ways: 1) to find moments of interaction (MoI) which we show are conducive to better representation learning and 2) as part of our audible state change loss (AStC) which encourages the model to develop state-aware representations.
We are glad the reviewers found that the proposed method is **novel and effective** (Reviewer wUoy), our evaluation **on two very relevant well-known ego centric benchmarks…makes the paper stronger** (Reviewer coBj) and that the **paper is clear in presentation and has provided an interesting view to self-supervised multi-modal representation learning in egocentric videos** (Reviewer WY3G). We summarize the clarification related to MoI and AStC below and further address reviewers’ individual feedback in detail by replying to their comment threads individually.

**1) Audible State Change Loss (AStC)**: AStC is designed to make the model learn the temporal change in visual state by increasing the probability of associating the forward (correct) direction of change ($\Delta v_t^{\text{frwd}}$) with the audio (Equation 4) and reducing the probability of associating of the backward (incorrect) direction of change ($\Delta v_t^{\text{bkwd}}$) with the audio (Equation 5). We mention this definition in Lines 158-161. By directly working on the direction of visual state change ($\Delta x$) rather than on the visual states ($x$), AStC allows the model to be more aware about the transitions in visual states over the course of any action or interaction in the environment.

**2) Moment of Interaction**: Referring to our definition of MoI in the paper (Lines 53-54, Lines 135-137), we intend to convey that an MoI can be any form of action or interaction among the entities in the environment that leads to a perceptible visual state change. Thus, MoIs are not restricted to just human-object and object-object interactions and can be associated with any form of interaction within the environment. We hypothesize that whenever there is such a visual state change, there is a high probability for it to be accompanied with a distinct/characteristic audio pattern (Lines 53-54) leading to an MoI detection. Such a formulation of MoI enables our method to focus on the timestamps in the video that provide relatively richer audio-visual signals for the model to learn meaningful feature representations that understand both visual states and temporal change in visual states. Note that we purposefully intend to detect generic enough MoIs to accommodate the unconstrained and unlabeled nature of the training data, and prevent being restricted to a closed set of human interactions.

We now answer each reviewer on a separate thread. We encourage reviewers to reach out during the discussion phase, if you still have questions. We’ll be prompt in our responses.

---

### Author Response · Authors · 2022-08-07
**Thanks and request for discussion**

Dear reviewers,

We thank you once again for the thorough consideration of our work. With the rebuttal/discussion period nearing a close, we would like to hear from reviewers on whether our rebuttal addressed your concerns. We are eager to clarify any remaining questions, help quell any remaining concerns, and more importantly, look for ways to improve the paper overall.

---

### Meta-Review · Area_Chair_REuf · 2022-08-30

**Recommendation:** Accept
**Confidence:** Certain

**Metareview:**

The paper presents self-supervised representation learning from egocentric video data. The reviewers unanimously support the paper. Although WY3G has not updated the rating, the reviewer commented that s/he is upgrading the rating to Weak Accept, making the paper get three unanimous Weak Accept ratings. All three reviewers find the idea of using audio to identify the state changes for learning audio-visual correlation interesting. The authors are encouraged to include added experimental results they provided during the discussion phase to the final version of the paper.


**Award:**

No

---

### Decision · Program_Chairs · 2022-09-14

Accept